# Differences in the Correlation between the Built Environment and Walking, Moderate, and Vigorous Physical Activity among the Elderly in Low- and High-Income Areas

**DOI:** 10.3390/ijerph19031894

**Published:** 2022-02-08

**Authors:** Peng Zang, Fei Xian, Hualong Qiu, Shifa Ma, Hongxu Guo, Mengrui Wang, Linchuan Yang

**Affiliations:** 1Department of Architecture and Urban Planning, Guangdong University of Technology, 729 Dongfeng E Rd, Guangzhou 510006, China; kenxin8989@163.com (P.Z.); 18308463173@163.com (F.X.); bkj338@163.com (H.Q.); guohx@163.com (H.G.); dream.pistil@163.com (M.W.); 2Department of Urban and Rural Planning, School of Architecture, Southwest Jiaotong University, Chengdu 611756, China; yanglc0125@swjtu.edu.cn

**Keywords:** physical activity, built environment, physical environment, the elderly, Guangzhou

## Abstract

Studies have proved that activity and fitness behaviors are closely related to the quality of life and health status of the elderly. However, different intensities of physical activity (PA)—walking, moderate PA, and vigorous PA—have different correlations with the built environment (BE). This study combines the high and low socioeconomic status (SES) of Guangzhou to establish two types of BE models. The physical activity time of 600 elderly people was collected from questionnaires. Through ArcGIS software, 300 m, 500 m, 800 m, and 1000 m buffer zones were identified, and the land use diversity, street design, population density, distance to destination, distance to public transportation—the five Ds of the BE—were measured. SPSS software was adopted to analyze the correlation between the BE and PA. Results: The PA of people living in low-SES areas was more dependent on the BE, whereas the correlation may be limited in high SES areas. Moreover, in low SES areas, walking was negatively correlated with street connectivity; moderate PA was positively correlated with street connectivity and the shortest distance to the subway station, but negatively correlated with the density of entertainment points of interest (POIs). Studying the relevant factors of the environment can propose better strategies to improve the initiative of the elderly to engage in PA.

## 1. Introduction

The rapid development of the social economy has brought people a high quality of living standard. The progress of science and technology has improved medical conditions, which have lengthened the average life span of the Chinese population. According to China’s seventh national population census conducted in 2020, the elderly population aged 60 years or above is 264.02 million, accounting for 18.7% (among which the population aged 65 years or above is 190.64 million, accounting for 13.5%). According to the international standard, a region is considered to have entered an aging society when its population aged 65 years or above reaches 7% and an aged society when the figure reaches 14%. This means that China is very close to a moderately aging society. According to the Prediction of China’s Health Commission, the elderly population in China is expected to peak at 487 million around 2050, accounting for 34.98% of the total population. Both the number of elderly people and the proportion of the total population have reached their peak, and China will be one of the countries with a relatively high degree of aging in the world (http://english.www.gov.cn/news/) (accessed on 27 December 2021). China will face severe challenges of population aging in the future. Therefore, the health problems of the aging population in China are also worthy of attention.

Guangzhou, the capital of Guangdong province, is a megacity. It is an important central city in China, an international business and trade center, and a comprehensive transportation center. As of 2021, the city administered 11 districts with a total area of 7434.40 square kilometers. The permanent population was 18.68 million, and the regional GDP was 2823.20 billion CNY. Guangzhou entered the aging society earlier than most cities in China in 1992. Since then, the degree of aging in Guangzhou has always been the highest in China, with a large number of elderly people and a high degree of aging. As of the end of 2019, there were 1.76 million people aged 60 and above in Guangzhou, accounting for 18.4 percent of the registered population, according to the “2019 Guangzhou Elderly Career Development Report and Elderly Population Data Booklet” by the Guangzhou Municipal Committee on Aging (http://www.gz.gov.cn/guangzhouinternational/index.html) (27 December 2021).

### 1.1. Benefits of Physical Activity

Physical activity is essential for older people, who belong to the most inactive population group. They have much to gain from increasing physical activity levels [1,2,3,4]. Despite the decline in physical function in older people, different levels of physical activity have different physical and mental health benefits. According to the World Health Organization’s physical activity recommendations for the elderly over 65, they should engage in at least 150–300 min of moderate-intensity aerobic activities per week, or at least 75–150 min of vigorous-intensity aerobic activities. The combination of moderate-intensity and vigorous-intensity physical activity can achieve huge health benefits. At the same time, sitting time still should be limited, and a variety of physical activities should be included in each week’s physical activity, focusing on the functional balance and strength training of moderate or higher intensity.

According to research, walking is moderate in intensity and easy to achieve, making it one of the most common and beneficial physical activities for the elderly [5,6,7]. Additionally, walking has also become the common travel mode adopted by the elderly [8]. Scholars in related research fields in the west began to pay attention to the walking ability, activity characteristics, and the influence mechanism of the elderly at a relatively early time. They found that the behavioral design of walking for older people was based on individual social characteristics [9,10]. Additionally, walking is the most important and direct way for the elderly to contact the city and the community [11]. The advantages of the health, convenience, and economy of walking are of positive significance for their physical and mental healing, as well as their sense of happiness in life and community identity [12,13,14,15,16,17].

The World Health Organization emphasizes that moderate physical activity is also suitable for older adults, and recommends it as an exercise option for older adults. Moderate physical activity includes brisk walking, community greening, cleaning, cycling, swimming, etc. [18,19]. Additionally, studies have demonstrated the importance of moderate physical activity in promoting health and preventing chronic diseases in older adults [18,20,21,22,23,24]. In addition, other studies have shown that some open spaces, such as parks, swimming pools, coastal roads, ice rinks, and some basic public facilities, are positively correlated with exercise time [25,26,27,28], and they could promote physical exercise of the elderly and reduce the health risks caused by the sedentary behaviors of the elderly [24,29,30].

Relatively, some vigorous physical activity (weight lifting, aerobics, jogging, fast cycling, etc.) may be difficult for most older adults, but vigorous physical activity can prevent frailty and cognitive impairment [18,24,28]. Some studies have confirmed that vigorous physical activity can inhibit sedentary time and regulate the physical function of the elderly, and promote travel in the elderly [31,32,33]. Although studies in this area are very rare, they are very important. Studies have shown that the accessibility of golf courses and tennis courts is positively correlated with physical activity [19].

A systematic review revealed that most studies have shown that walking is the most common and most appropriate physical activity, as well as the impact of moderate and vigorous physical activity on the health of the elderly. Additionally, nowadays, the physical fitness of the elderly is becoming better and better, and the combination of walking, moderate, and vigorous physical activity may be more beneficial.

### 1.2. The Influence of Built Environment on Physical Activity

The built environment is considered to be particularly relevant to the PA of the elderly [34,35,36]. Studies have shown that the important safeguard for elderly groups to participate in physical activity and guarantee for the elderly to integrate into social life is the built environment [37,38]. The built environment can promote or inhibit the physical activity of the elderly [28,39,40]. Studies have shown that land mix and street connectivity, two built environment attributes, are related to physical activity for the elderly [26,41,42]. Street connectivity is significantly related to the number of pedestrian blocks [43,44]. The higher the walkability, the longer time the elderly spend for moderate and vigorous PA [22,28,40]. Elderly people are afraid to travel on streets with obstacles, roads with steep slopes, and roads with many intersections [28,45]. Traffic injuries are an especially worrying limiting factor of outdoor physical activity [28,46].

A large number of studies have shown that the neighborhood walkability [47,48,49], accessibility (e.g., commercial, education, public transportation stations, parks, green spaces, and leisure facilities) [5,50,51], attractiveness (green space, and parks) [52,53], and other objective built environment characteristics may support or hinder the general, transport, and recreational walking behavior of the elderly.

In addition, the elderly’s perception of characteristics of the built environment in the neighborhood also has an effect on activities. For example, the perception of walking facilities [7,54,55], walkability [51], accessibility [54,55], and safety [50,54,55] may significantly affect the general, transport, and recreational walking participation and activity volume. In addition, socioeconomic attributes such as age, education level, SES, and health status are also important influencing factors [51,56,57].

### 1.3. Thrust of Our Research

Guangzhou has experienced a rapid urbanization and urban expansion, so its demographic structure and socioeconomic status are vastly different from other cities. Therefore, exploring the impact of the built environment on physical activity in such a context may be a new discovery. For example, different SESs may affect the correlation between the two.

We closely follow the background of the aging population and prominent health problems of the elderly and focus on the research theme of daily walking and physical activity. The people-oriented concept is integrated into the research context, and the influence of the urban built environment on walking activities of the elderly is explored, as well as the optimization and adjustment of urban space design.

In this study, two types of regions were divided according to different socioeconomic levels of Guangzhou. Then, the differences in walking, moderate, and vigorous physical activity among the elderly were compared in combination with the population density at the high, middle, and low levels. This study uses the questionnaire to measure the physical activity time of older people. ArcGIS technology and Statistical Products and Services solution (SPSS) (Scientific Software International, Inc., Skokie, IL, USA) software were used to determine and calculate the relevant differences between different buffer zones and different built environments (land mixing degree, street connectivity, etc.) and physical activity of the elderly with different intensities. For a representative figure of the study protocol, see Figure 1.

## 2. Materials and Methods

### 2.1. Selection of Research Objects and Research Areas

Based on the International Physical Activity Questionnaire—Long Version (IPAQ-LC)—, 600 elderly people over 65 years old were selected for the questionnaire survey in this study. The questionnaire also included the occupation, age, educational background, and other information of the research object. The housing price of Guangzhou was divided into 10 groups, and the highest group and the lowest group were removed. The remaining groups were classified as the high SES group (30,000–55,000 RMB/m^2^) and the low SES group (10,000–30,000 RMB/m^2^). Then, the population density of Guangzhou was taken as one of the dividing conditions. The city was finally divided into two groups of SES districts and six combinations, as shown in Table 1. Then, 12 sample communities were selected from the six regions.

### 2.2. Built Environment Data

ArcGIS was used to analyze the 5 Ds of the built environment (the 5 Ds variables in this study included land use diversity, street connectivity (design), population density, distance to the destination, distance to public transportation) of selected sample communities. The sDNA (sDNA is the world’s leading 2D and 3D spatial network analysis software for GIS, CAD, command line and Python, using industry standard network representations; we calculated accessibility and predicted flows of pedestrians, cyclists, vehicles, and public transport users; these informed models of health, community cohesion, land values, town center vitality, land use, accidents, and crime; We offered a simpler alternative to transport modeling, particularly for sustainable transport) tool in ArcGIS was used to calculate the proximity centrality (the proximity centrality: respond to the importance of the global location) and the intermediate centrality (the intermediate centrality: respond to the center of gravity of a small area, i.e., the density of the network) of 300 m, 500 m, 800 m, and 1000 m [58], and conducted a comparative analysis of physical activity.

The built environment measurement data came from the ArcGIS database of Map Heaven and Earth, as well as the data obtained by field investigation and measurement of our research team. Based on a review of relevant studies [44,59], there were seven important variables in affecting the physical activity of the elderly. They were: (1) population density, which considers the number of people living on the land per unit area, measured in ten thousand people per square kilometer; (2) land use diversity, which considers six types of land use: residential, commercial, educational, medical, recreational, and service; its calculation [60] method was as follows:(1)$$H=-(\overset{n}{\underset{i=1}{\sum }}P_{i}*\ln\left(P_{i}\right))/\ln\left(n\right)$$
where H represents the land use mix score ranging from 0 (a complete advantage of one type of land use) to 1 (equal distribution of all types of land use), P_i_ is the proportion of the i-th land use, and n is the number of land use types. A higher score represented a greater degree of mixed land use. (3) Street connectivity; (4) the number of bus stops; (5) the number of subway stations; (6) distance to the nearest bus/subway station; the network distance was calculated in ArcGIS; (7) points of interests (POIs) density, which was measured at the residence community level through ArcGIS.

### 2.3. Physical Activity Data

The walking time and physical activity time of the elderly in one week were collected and analyzed through a questionnaire survey of the elderly in various communities in Guangzhou. The questionnaire used was the International Physical Activity Questionnaire —Long Version (IPAQ-LC). There were 600 questionnaires (597 valid in total), plus 20 in-depth interview data and recording files. We mainly collected sitting, walking, moderate physical activity MPA), and vigorous physical activity (VPA) times.

### 2.4. Data Analysis

First, we classified and sorted the physical activity data of the elderly (≥65 years old) and the degree of land blending and street connectivity calculated by ArcGIS for the relevant building environment. Then, we used SPSS (Statistical Products and Services solution) software to carry out correlation analysis to detect the differences in the correlation between the same degree of physical activity and the built environment in different built environments, and the correlation differences between the same degree of physical activity and the built environment in the same built environment. When performing correlation analysis, we used linear regression model testing (Pearson correlation model and stepwise regression model), which proved that the data had no collinearity through VIF < 5, used a 95% confidence interval (CI) to evaluate the significance of all variables, and evaluated the significance of all variables. Buffer sizes (300 m, 500 m, 800 m, and 1000 m) were all subjected to correlation analysis.

## 3. Results

After the final data analysis of the questionnaires, three invalid questionnaires were excluded. The 597 valid questionnaires were then subjected to a correlation analysis. The scope of the questionnaire was a total of twelve communities in the two categories of built-up environments that were classified in the previous period; the questionnaire was aimed at people aged 65 and over. The built environment was compared by high-income and low-income areas, which included the land mix, street connectivity, population density, and buffer scale; the physical activity level was divided into four categories: sitting, walking, moderate physical activity, and vigorous physical activity.

### 3.1. Descriptive Statistics

It was clear that walking, moderate, and vigorous physical activity times were generally higher in high socioeconomic areas than in low socioeconomic areas, and, conversely, the sedentary time was generally higher for older people in low socioeconomic areas than in high socioeconomic areas. Older people’s exercise mainly focused on walking and moderate physical activity, with the most time spent on moderate physical activity.

The land use mix, intersection density, the number of bus stops, the number of metro stations, and overpasses all increased with the socioeconomic level, while street connectivity decreased with an increasing socioeconomic level.

According to the data in Table 2, in the low-income economic zone, older people spent 330.02 min/week sitting still, 387.44 min/week walking, 522.29 min/week performing moderate PA, and 104.20 min/week performing vigorous PA. In total, 92.3% of seniors would perform moderate PA and 22% would perform vigorous PA. In the high-income-area economic zone, seniors spent 307.54 min/week sitting still, 461.12 min/week walking, 712.18 min/week performing moderate physical activity, and 188.35 min/week performing vigorous physical activity. A total of 80.13% chose moderate PA, and 20.20% performed vigorous PA.

### 3.2. Correlation Analysis

At the low socioeconomic level, the physical activity of the elderly was influenced by the built environment and other factors. The density of commercial POIs, entertainment POIs, medical POIs, education POIs, public administration POIs, and the number of bus stations were all positively correlated with the sedentary time of the elderly, but negatively correlated with the proximity centrality of 800 m, proximity centrality of 1000 m, and the distance to the subway station. The walking time was positively correlated with the intermediate centrality of 500 m, 800 m, and 1000 m, while negatively correlated with street connectivity and the number of subway stations. The moderate PA of the elderly was positively correlated with street connectivity, a proximity centrality of 500 m, intermediary centrality of d500 m, and distance to the subway station, but negatively correlated with the density of entertainment POIs and the number of subway stations and overpasses. In contrast, there was no correlation between vigorous PA and built environment in the elderly.

According to Table 3, under the low SES, only in the 800 m and 1000 m buffer zones, the sedentary time was positively correlated with the intersection density (R = 0.132 *, *p* = 0.023; R = 0.137 *, *p* = 0.018). At the same time, the sedentary time was negatively correlated with a proximity centrality of 800 m (R = −0.135 *, *p*=0.020) and proximity centrality of 1000 m (R = −0.134 *, *p* = 0.021). The distance to the subway station (R = −0.222 *, *p* = 0.025) was negatively correlated with the sedentary time only in the 800 m buffer zone but not in the 1000 m buffer zone.

The walking time was negatively correlated with street connectivity in the buffer zone of 500 m (R = −0.172 *, *p* = 0.011) and 800 m (R = −0.137 *, *p* = 0.044). The walking time was positively correlated with the mediating centrality of 500 m (R = 0.016 *, *p* = 0.016), 800 m (R = 0.209 **, *p* = 0.002), and 1000 m (R = 0.172 *, *p* = 0.011), and only in the buffer zone of 800 m, the shortest distance to the subway station (R = −0.250 *, *p* = 0.036) was negatively correlated with the walking time, and the other buffers had no correlation.

The moderate PA time for the elderly was positively correlated with street connectivity (R = 0.149 *, *p* = 0.025) only within the 300 m buffer zone, positively correlated with the distance of the subway station (R = 0.333 **, *p* = 0.003) in the 800 m buffer zone, but negatively correlated with the number of subway stations in the 800 m and 1000 m buffer zones (R = −0.281 *, *p* = 0.013; R = −0.217 *, *p* = 0.020); moderate PA was positively correlated with a proximity centrality of 500 m (R = 0.141 *, *p* = 0.034) and intermediate centrality of 500 m (R = 0.133 *, *p* = 0.045). In the buffer zones of 500 m (R = −0.211 *, *p* = 0.024), 800 m (R = 0.201, *p* = 0.032), and 1000 m (R = −0.200 *, *p* = 0.034), the moderate PA time was negatively correlated with the number of overpasses. Vigorous PA was found not so relevant, so the correlation values for VPA were removed from Table 3.

According to Table 4, at a high SES, the time of walking, moderate PA, and vigorous PA was not found to have any correlation with the built environment, but only the sedentary time was found to be correlated with the built environment, where the sedentary time was positively correlated with the number of subway stations within 300 m and 500 m buffer zones (R = 0.200 *, *p* = 0.046; R = 0.200 *, *p* = 0.046), a proximity centrality of 500 m (R = 0.127 *, *p* = 0.029), and the distance to the nearest subway station being within 300 m and 500 m buffer zones (R = −0.200 *, *p* = 0.029), and the number of overpasses in the 300 m and 1000 m buffer zones (R = −0.200 *, *p* = 0.046; R = −0.139 *, *p* = 0.050) were negatively correlated; no correlation was found between the walking time and other built environments.

### 3.3. Regression Analysis

Table 5 is the summary and collation of the stepwise regression analysis of sitting, walking, moderate, and vigorous physical activity at high and low SES.

At low SES, neither the sedentary time nor vigorous PA time had a correlation with the built environment. The walking time was negatively correlated with street connectivity in the 800 m buffer zone (*t* = −2.400, *p* = 0.019 *) and intersection density in the 1000 m buffer zone (*t* = −2.418, *p* = 0.018 *). In the 800 m and 1000 m buffer zones, the moderate PA time was positively correlated with the proximity centrality of 800 m (*t* = 3.140, *p* = 0.002 **; *t* = 3.139, *p* = 0.002 **).

At the high socioeconomic level, only the sedentary time was negatively correlated with the density of recreational POIs (*t* = −2.020, *p* = 0.046 *) in the 300 m buffer zone. Street connectivity in the 500 m buffer zone (*t* = 2.117, *p* = 0.037 *), proximity centrality in the 800 m buffer zone (*t* = 2.146, *p* = 0.034 *), and the number of broken roads in the 1000 m buffer zone were positively correlated with the sedentary time.

## 4. Discussion

This study mainly explored the correlation and consistency between the built environment characteristics in different scales of buffer zones divided by ArcGIS and the sedentary time, walking time, moderate PA time, and vigorous PA time of the elderly according to their high or low SES. Among them, groups were grouped according to the socioeconomic level and the 300 m, 500 m, 800 m, and 1000 m buffers were derived using the ArcGIS tool. In general, there was no correlation between the self-rated health and physical activity regardless of the income level. In this study, however, the self-rated health was negatively correlated with physical activity regardless of the socioeconomic level. Under the low SES, the sedentary time, walking time, and moderate PA time of the elderly were differently correlated with the built environment. Street connectivity, the number of overpasses, and the number of subway stations all affected the physical activity needs of the elderly.

This study showed that at a low SES, the walking time was negatively correlated with street connectivity, which was quite different from previous studies [61], which suggested that the higher the street connectivity, the better the mobility of the elderly. However, this study also found that the mediated centrality of the street was positively correlated with walking and moderate PA, which was inconsistent with the above research results of street connectivity. The reason may be that street connectivity needs to be considered in conjunction with mediated centrality to better assess the correlation between the street environment and physical activity in the elderly. This result may be explained by the low SES in Guangzhou, where the street infrastructure needs to be improved, and although the street connectivity was high, it was not suitable for the elderly to walk. Compared with existing studies [62,63], older people at a low SES did not perceive the benefits of a built environment conducive to physical activity, which also explained the high variability in the data that arose from Table 2. There was also a negative correlation between the walking time and the number of subway stations. Although the subway has been an increasingly popular means of travel in recent years, its design is more suitable for young people, and subway design is still not friendly to the elderly [64]. Previous studies have confirmed that older people are more likely to be physically active for transport purposes in communities suitable for walking [65]. Similarly, in this study, we found that the shortest distance to the subway station reduced the walking time of the elderly. Perhaps older people are more likely to take the subway than spend time walking when the design of the subway allows. This indicated that, in the buffer zone of 1000 m, the shortest distance of the subway station was not related to this, and walking over a long distance and time exceeded the walking intention of the elderly.

For the low SES, moderate PA was positively correlated with the shortest subway station distance, which was consistent with the results of current studies [66]. Assuming that the elderly are willing to carry out moderate PA, the shortest subway station distance would promote their further exercise. However, when the number of overpasses with stairs increased, the moderate PA of the elderly decreased. The study found that the number of overpasses was generally negatively correlated with walking, which was the same as previous studies [45]. The elderly preferred streets with sidewalks and continuous existence, which would reduce their willingness to perform physical activity. When the elderly faced steep slopes and stairs, their willingness to travel would decrease. Entertainment POIs were also negatively correlated with the moderate PA time, and in a low SES, expenditure is indeed a big problem.

However, for the low SES, vigorous physical activity was difficult for the elderly to accept due to physical factors, so these elderlies rarely chose vigorous PA. Therefore, in this study, there was almost no correlation between vigorous PA and the built environment, which was similar to previous studies [27]. In addition, during the preliminary research period, the author conducted a questionnaire survey on the elderly and found that an elderly person often carried out vigorous physical activity, such as cycling around the Pearl River, morning running and night running in the park, etc., but most elderly people would not choose this kind of vigorous PA.

For the high SES, this study found that there was no correlation between walking, moderate PA, and vigorous PA with the characteristics of the built environment. Lots of literature mainly studies the correlation between the built environment and physical activity [67,68] and supports this relationship, but according to ArcGIS and other measures, for neither SES, the elderly could obtain the same benefits from the built environment of “fitting physical activity” [27,69].

China’s current pension policy has been in place for a long time, based on the country’s national conditions. However, due to a large number of the aging population, complicated historical circumstances, etc., an adequate policy articulation and adjustments are still needed. Spontaneous physical activity for the elderly is encouraged in the community; it is recommended that the design of the elderly aspect be emphasized in urban planning, as well as healthy aging for every elderly person when conditions permit.

## 5. Conclusions

This study was conducted in Guangzhou, China, and the results showed that street connectivity and the shortest subway station distance would reduce the elderly’s demand for walking in a low SES, but could promote the elderly to carry out moderate PA. Moreover, the built environment characteristics (e.g., the number of subway stations and the number of overpasses) had different effects on the physical activity needs of the elderly in different buffer zones. For the high SES, there was little correlation between the built environment and the physical activity of the elderly.

Results from the same type of study showed that in high-SES countries, there was no match between the physical activity and objectively measured built environment characteristics under cross-sectional, independent variables that have previously been associated with physical activity Our research especially took into account the social contradictions under the rapid development facing Chinese society. This study suggested that people living in low-income settings may not experience all the benefits of living in age-friendly physical activity communities. Additionally, physical activity is utilitarian rather than voluntary when taking into account the physical activity limitations and differences in the built environment characteristics of people at lower socioeconomic levels. Then, the study of environmental factors could put forward better strategies to improve the enthusiasm of the elderly in the low socioeconomic environment to carry out spontaneous physical activity. The link between the physical activity of older people and built environment characteristics needs to be further explored in the wider Chinese region, with the aim of creating more comfortable and safer communities with the built environment.

## Figures and Tables

**Figure 1 ijerph-19-01894-f001:**
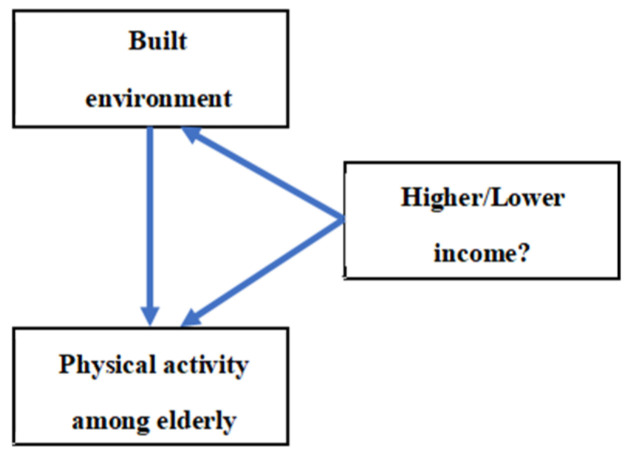
Study protocol.

**Table 1 ijerph-19-01894-t001:** Combination of study areas.

	Low-Density	Medium-Density	High-Density
Low-income	Li Cheng Central District Huadong Town	Chia Cao He Bin North Road	Dong Long Osmanthus Valley
High-income	Han Xi Agile	Chang Xing Tianhe Park	Hang Kou Guang Gang New Town

**Table 2 ijerph-19-01894-t002:** Descriptive statistics of population characteristics and built environment at high SES and low SES.

	Low SES			High SES		
	Percentage	Mean	Standard Deviation	Percentage	Mean	Standard Deviation
Total	50.25			49.75		
Age						
65–74.	88.70			84.80		
75–84.	10.30			13.10		
85+	1.00			2.00		
Education level						
Medium and above	10.30			7.10		
Elementary education	83.70			74.70		
None	6.00			18.20		
Self-assessed health						
Good	19.00			22.20		
General	78.70			72.10		
Bad	2.30			5.70		
Population density		1.17	1.31		1.30	0.66
Sitting		330.02	274.33		307.54	240.97
Walking		387.44	628.29		461.12	624.42
MPA		522.29	907.95		712.18	1794.85
VPA		104.20	328.83		188.35	807.98
Land use diversity		0.43	0.21		0.48	0.10
Street connectivity		1.77	0.19		1.71	0.18
Intersection density		92.72	59.73		120.09	38.16
Proximity centrality 300 m		0.48	0.08		0.46	0.15
Proximity centrality 500 m		0.54	0.05		0.53	0.04
Proximity centrality 800 m		0.51	0.05		0.51	0.07
Proximity centrality 1000 m		0.51	0.07		0.51	0.06
Intermediate centrality 300 m		0.14	0.04		0.13	0.03
Intermediate centrality 500 m		0.13	0.02		0.13	0.04
Intermediate centrality 800 m		0.13	0.04		0.12	0.05
Intermediate centrality 1000 m		0.14	0.04		0.12	0.03
Commercial POIs		120.62	105.69		154.71	104.07
Recreational POIs		14.72	11.75		23.12	14.39
Medical POIs		19.62	18.21		19.58	11.06
Education POIs		27.06	24.05		45.98	43.88
Public Administration POIs		43.42	42.04		35.84	22.42
Number of bus stops		18.60	15.81		21.38	10.95
Bus station distance		222.52	193.45		195.75	199.58
Number of subway stations		4.53	8.01		6.88	3.30
Distance to subway stations		424.88	493.27		784.61	409.17
Number of overpasses		1.17	1.78		1.51	1.71
Number of broken roads		93.51	37.44		113.65	46.72

**Table 3 ijerph-19-01894-t003:** Pearson correlation analysis under low SES.

		300 m Buffer Zone			500 m Buffer Zone			800 m Buffer Zone			1000 m Buffer Zone		
		sitting	walking	MPA	sitting	walking	MPA	sitting	walking	MPA	sitting	walking	MPA
Self-report health	R	0.088	−0.270 **	0.117	0.088	−0.270 **	0.117	0.088	−0.270 **	0.117	0.088	−0.270 **	0.117
	*p*-value	0.129	0.000	0.080	0.129	0.000	0.080	0.129	0.000	0.080	0.129	0.000	0.080
Street connectivity	R	0.049	0.020	0.149 *	0.024	− 0.172 *	0.095	0.011	−137 *	0.098	0.004	0.123	0.081
	*p*-value	0.397	0.766	0.025	0.674	0.011	0.153	0.854	0.044	0.142	0.949	0.072	0.225
Intersection density	R	0.084	0.020	0.062	0.099	0.003	0.002	0.132 *	0.039	0.014	0.137 *	0.052	0.015
	*p*-value	0.148	0.765	0.349	0.087	0.970	0.971	0.023	0.563	0.829	0.018	0.448	0.823
Proximity centrality 500 m	R				0.059	0.079	0.141 *	0.059	0.079	0.141 *	0.061	0.079	0.138 *
	*p*-value				0.310	0.247	0.034	0.310	0.247	0.034	0.293	0.249	0.038
Proximity centrality 800 m	R							−0.135 *	0.004	0.067	−0.138 *	0.003	0.063
	*p*-value							0.020	0.952	0.312	0.017	0.970	0.345
Proximity centrality 1000 m	R										−0.134 *	0.041	0.077
	*p*-value										0.021	0.547	0.249
Intermediate centrality 500 m	R				0.057	0.161 *	0.133 *	0.057	0.161 *	0.133 *	0.057	0.163 *	0.131 *
	*p*-value				0.328	0.018	0.045	0.328	0.018	0.045	0.323	0.016	0.049
Intermediate centrality 800 m	R							0.061	0.209 **	0.111	0.060	0.209 **	0.112
	*p*-value							0.293	0.002	0.094	0.299	0.002	0.093
Intermediate centrality 1000 m	R										0.049	0.172 *	0.094
	*p*-value										0.395	0.011	0.159
Commercial POIs	R	0.156 *	0.089	0.088	0.172 **	0.082	0.043	0.163 **	0.079	0.071	0.160 **	0.084	0.089
	*p*-value	0.014	0.226	0.224	0.003	0.227	0.521	0.005	0.247	0.288	0.006	0.218	0.184
Recreational POIs	R	0.133 *	0.106	0.072	0.146 *	0.128	0.140	0.140 *	0.125	−160 *	0.137 *	0.066	0.103
	*p*-value	0.037	0.150	0.315	0.021	0.082	0.051	0.027	0.090	0.026	0.018	0.335	0.122
Medical POIs	R	0.116	0.067	0.005	0.123	0.057	0.007	0.144 *	0.074	0.067	0.167 **	0.071	0.039
	*p*-value	0.068	0.364	0.940	0.053	0.436	0.918	0.023	0.318	0.356	0.004	0.296	0.555
Education POIs	R	0.152 **	0.082	0.019	0.152.*	0.069	0.002	0.163 **	0.080	0.059	0.164 **	0.078	0.064
	*p*-value	0.008	0.230	0.772	0.008	0.313	0.980	0.005	0.240	0.379	0.004	0.256	0.337
Public administration POIs	R	0.144 *	0.082	0.046	0.160 **	0.054	0.014	0.157 **	0.052	0.035	0.165 **	0.070	0.050
	*p*-value	0.023	0.268	0.521	0.006	0.427	0.836	0.007	0.445	0.599	0.004	0.307	0.458
Number of bus stops	R	0.167 *	0.132	0.021	0.154 *	0.016	0.033	0.160 *	0.010	0.042	0.163 **	0.024	0.046
	*p*-value	0.018	0.111	0.793	0.015	0.829	0.657	0.012	0.892	0.570	0.010	0.752	0.527
Number of subway stations	R	.c	.c	.c	.c	.c	.c	0.166	−0.250 *	−0.281 *	0.092	0.121	−0.217 *
	*p*-value							0.094	0.036	0.013	0.260	0.211	0.020
Distance to subway stations	R	.c	.c	.c	.c	.c	.c	−0.222 *	0.153	0.333 **	0.027	0.138	0.042
	*p*-value							0.025	0.203	0.003	0.744	0.154	0.654
Number of overpasses	R	.c	.c	.c	0.050	0.103	−0.211 *	0.043	0.113	−0.201 *	0.041	0.115	−0.200 *
	*p*-value				0.546	0.280	0.024	0.601	0.236	0.032	0.617	0.231	0.034

*. Correlation is significant at the 0.05 level (2-tailed). **. Correlation is significant at the 0.01 level (2-tailed).

**Table 4 ijerph-19-01894-t004:** Pearson correlation analysis at high SES.

		300 m Buffer Zone		500 m Buffer Zone		800 m Buffer Zone		1000 m Buffer Zone	
		sitting	walking	sitting	walking	sitting	walking	sitting	walking
Self-report health	R	0.028	−0.160 *	0.028	−0.160 *	0.028	−0.160 *	0.028	−0.160 *
	*p*-value	0.637	0.015	0.637	0.015	0.637	0.015	0.637	0.015
Proximity centrality 500 m	R			0.127 *	0.001	0.127 *	0.001	0.127 *	0.000
	*p*-value			0.029	0.990	0.029	0.990	0.030	0.997
Number of subway stations	R	0.200 *	0.127	0.200 *	0.127	0.002	0.108	0.029	0.075
	*p*-value	0.046	0.260	0.046	0.260	0.970	0.138	0.616	0.254
Distance to subway stations	R	−0.200 *	0.127	−0.200 *	0.127	0.030	0.038	0.039	0.055
	*p*-value	0.046	0.260	0.046	0.260	0.637	0.606	0.505	0.405
Number of overpasses	R	−0.200 *	0.127	0.134	0.083	0.139	0.052	−0.139 *	0.061
	*p*-value	0.046	0.260	0.058	0.297	0.050	0.513	0.050	0.445

*. Correlation is significant at the 0.05 level (2-tailed).

**Table 5 ijerph-19-01894-t005:** Stepwise regression analysis of sitting, walking, moderate, and vigorous physical activity at low and high socioeconomic levels.

In a 500 m buffer zone at low socioeconomic level								
	Sitting		Walking		MPA		VPA	
	Beta (95% CI)	*p*-value	Beta (95% CI)	*p*-value	Beta (95% CI)	*p*-value	Beta (95% CI)	*p*-value
Age	0.105	0.411	0.084	0.656			0.704	0.034
Education level	0.365	0.009	0.009	0.964			0.245	0.411
Self-report health	0.282	0.041	0.306	0.095			0.120	0.694
In an 800 m buffer zone at low socioeconomic level								
Age							0.542	0.030
Education level	0.417	0.000			0.179	0.112	1.327	0.207
Street connectivity			0.283	0.019 *	0.238	0.812		
Proximity centrality 800 m	0.166	0.074	0.414	0.680	0.347	0.002 **	1.130	0.279
In a 1000 m buffer zone at low socioeconomic level								
Age					0.158	0.148	0.542	0.030
Education level	0.417	0.000			0.192	0.087		
Intersection density	1.018	0.311	0.285	0.018 *	0.034	0.805	0.266	0.279
Proximity centrality 800 m	0.166	0.074			0.347	0.002 **		
In a 300 m buffer zone at high socioeconomic level								
	Sitting		Walking		MPA		VPA	
	Beta (95% CI)	*p*-value	Beta (95% CI)	*p*-value	Beta (95% CI)	*p*-value	Beta (95% CI)	*p*-value
Age			0.201	0.075 *				
Self-report health	0.128	0.211					0.381	0.107
Recreational POIs	0.200	0.046 *	0.079	0.499			0.040	0.869
In a 500 m buffer zone at high socioeconomic level								
Age	0.122	0.230	0.201	0.075 *				
Self-report health	0.049	0.626					0.381	0.107
Street connectivity	0.209	0.037 *	0.092	0.424			0.040	0.869
In an 800 m buffer zone under high socioeconomic level								
Education level	0.319	0.095	0.237	0.010 *				
Proximity centrality 800 m	0.177	0.034 *						
In a 1000 m buffer zone under high socioeconomic level								
Population density			0.153	0.053				
Self-report health			0.068	0.384			0.270	0.141
Distance to bus station	0.059	0.536	0.025	0.755	0.124	0.114	0.063	0.733
Number of guillotines	0.141	0.046 *	0.068	0.478	0.014	0.892		

*. Correlation is significant at the 0.05 level (2-tailed). **. Correlation is significant at the 0.01 level (2-tailed).

## Data Availability

The date can be obtained from the corresponding author upon reasonable request.

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
