# Peer review of "Differences in the Correlation between the Built Environment and Walking, Moderate, and Vigorous Physical Activity among the Elderly in Low- and High-Income Areas"

_ijerph, 2022, doi:10.3390/ijerph19031894_

Round 1

Reviewer 1 Report

See file attached

Reviewer 2 Report

Summary

This paper presents an analysis of correlation between different levels of physical activity (PA) of elderly people (age>65) and different built environment (BE) characteristics. The performed analysis is based on the population of Guangzhou (also known internationally as Canton City), the capital of Guangdong Province in southern mainland China. Moreover, it is considered a distinction between low- and high-income areas. For elderly belonging to a high socio-economic status (SES) area, the presented results indicate a weak correlation between the physical activity and the built environment. Regarding elderly living in a low SES area, the results indicate that BE characteristics have different effects on the PA of the elderly according to different buffer zones.

Major comments

The readability of the paper must be substantially improved.

To help the authors I list important issues that must be addressed (this should not be seen as an exhaustive list):

  • The abstract contains several unclear sentences, e.g., lines 17-19: “The date of PA in the elderly was statistically analyzed by questionnaires (N=597). Through GIS to identify 300, 500, 800, and 1,000 m buffers and SPSS software to 18 analyze the correlation between the BE and PA of the elderly with different intensities in”. I suggest to rewrite carefully the abstract to convey to the readers, in a clear way, the real contribution of the paper.
  • All acronyms should be corrected defined, e.g., line 18: “GIS”; line 22: “POI”; Table 2: “MPA” and “VPA”.
  • Since the analysis is solely performed based on data from one city, it is important to further describe this city and its relevance to the present study. I suggest the authors to further elaborate in the paragraph that ends in line 50.
  • Line 90: “although” should be deleted.
  • The sentence between lines 97-99 should be rewritten to avoid the repletion of the word “guarantee”.
  • Along the text the authors use “elderly” and “older adults” as synonymous. Accordingly, in line 104: “adults” should be “elderly” or “older adults”.
  • The sentence between lines 130 and 134 is too long and hence hard to follow for the reader. Please break it.
  • The authors should avoid the repetitions of the same idea throughout the text. E.g., in line 145 it is said that “600 elderly people over 65 years old”. Nevertheless, it was made clear previously that, in this study, it is elderly a person over 65 years old. The “600” elderly people over 65 years old” entails a redundancy.
  • All the data analysis should be further explained. E.g., Line 190: “we used linear regression model testing, etc.” is not an acceptable sentence.
  • Typo In Table 2: “2..0” should be “2.0”.
  • The authors should be more careful in the interpretation of the results. E.g., line 295: “confirmed” should be replaced for “suggest”.
  • Based on the results obtained, I recommend the authors to complement the “Discussion” Section by discussing possible policies that can be undertaken to contribute to a better health and well-being of the elderly.

Other comments

The equation presented in line 168 should be numbered and explained. What is the reference that supports this equation?

Round 2

Reviewer 2 Report

I congratulate the authors on their improvement of the manuscript. Nevertheless, there are some minor issues that need to be addressed.

Comment #1

In this version there are still run-on sentences that make the text more difficult to read. See, e.g., lines 18-21 in the abstract: “Through ArcGIS software, 300m, 500m, 800m, and 1000m buffer zones were identified, and land use diversity, street design, population density, distance to destination, distance to public transportation - the 5Ds of the BE were measured, and SPSS software was adopted to analyze the correlation between the BE and PA.” This could be simplified to: “Through ArcGIS software, 300m, 500m, 800m, and 1000m buffer zones were identified, and land use diversity, street design, population density, distance to destination, distance to public transportation - the 5Ds of the BE were measured. SPSS software was adopted to analyze the correlation between the BE and PA.” The presentation should not be so laborious to read. Please avoid this kind of too long sentences throughout the text.

Comment #2

A Figure 1 was added in this version. However, I cannot find any reference to this figure in the text.

Comment #3

The descriptive statistics presented in Table 2 should be self-explained. What is the measure unit of the population density, sit, walking, MPA, VPA, etc.? Moreover, be careful that 300/597 and 297/597 are not percentages.

Other comments

Typos detected:

  • Line 25: “PoIs” should be “POIs.”
  • Lines 137-138: “Statistical Products and Services solution software (SPSS)…” should be “Statistical Product and Service Solutions (SPSS)….”
  • Line 170: delete the first “(” which is in excess.

I suggest proofreading the paper bear in mind the above comments.
